# Conversion of Secondary C3-C4 Aliphatic Alcohols on Carbon Nanotubes Consolidated by Spark Plasma Sintering

**DOI:** 10.3390/nano11020352

**Published:** 2021-02-01

**Authors:** Serguei Savilov, Evgeniya Suslova, Vsevolod Epishev, Evgeniya Tveritinova, Yuriy Zhitnev, Alexander Ulyanov, Konstantin Maslakov, Oksana Isaikina

**Affiliations:** 1Department of Chemistry, Lomonosov Moscow State University, Leninskie Gory, 119991 Moscow, Russia; savilov@chem.msu.ru (S.S.); olimpxumuk@gmail.com (V.E.); eatver@mail.ru (E.T.); zhitnev@kge.msu.ru (Y.Z.); a-ulyanov52@yandex.ru (A.U.); nonvitas@gmail.com (K.M.); hardi1@yandex.kz (O.I.); 2A.V. Topchiev Institute of Petrochemical Synthesis, Russian Academy of Sciences, Leninsky Prospekt, 29, 119991 Moscow, Russia

**Keywords:** carbon nanotubes, spark plasma sintering, gas-phase oxidation, catalyst, dehydration, dehydrogenation, activation energy, secondary alcohols, electronic paramagnetic resonance

## Abstract

We analyze how the changes in the dimension of carbon nanomaterial (CNM) affect their catalytic conversion of secondary aliphatic alcohols. Carbon nanotubes (CNTs) consolidated by spark plasma sintering (SPS) were inactive in the conversion of secondary C_3_-C_4_ aliphatic alcohols because of the «healing» of defects in carbon structure during SPS. Gas-phase treatment of consolidated CNTs with HNO_3_ vapors led to their surface oxidation without destruction of the bulk structure of pellets. The oxygen content in consolidated CNTs determined by X-ray photoelectron spectroscopy increased from 11.3 to 14.9 at. % with increasing the oxidation time from 3 to 6 h. Despite the decrease in the specific surface area, the oxidized samples showed enhanced catalytic activity in alcohol conversion because of the increased number of oxygen radicals with unpaired electrons, which was established by electron paramagnetic resonance spectroscopy. We conclude that the structure of CNM determines the content and/or ratio of *sp*^2^ and *sp*^3^-hybridized carbon atoms in the material. The experimental and literature data demonstrated that *sp*^3^-hybridized carbon atoms on the surface are probably the preferable site for catalytic conversion of alcohols.

## 1. Introduction

The size, porosity, specific surface area and functionalization degree of structured carbon nanomaterials (CNMs) can be tuned in a wide range by varying the synthesis conditions and/or post-synthetic processing. This makes them attractive for application as catalyst supports [1,2]. Even raw (unmodified) CNMs can catalyze several reactions [3,4]: oxidative dehydrogenation of ethylbenzene [5] and light alkanes [6,7,8,9], conversion of aliphatic alcohol [10,11,12,13,14] and so forth.

Secondary alcohol conversion is used in industry including fine chemical synthesis and can be considered as a test reaction for basic and/or acidic sites on the surface of carbon materials [15,16]. A wide range of carbon nanomaterials were tested as catalysts including nanodiamonds (NDs) and their oxidized and reduced forms [10,17,18], carbon nanofibers [12], carbon nanotubes (CNTs) with different morphology [12,13,14], oxidized CNTs [11,13], carbon nanoflakes (CNFs) and its nitrogen-doped derivatives [11,18], graphene oxide [19] and activated carbon [15,16,20,21].

There are several products of secondary alcohols conversion. Aldehydes and ketones are formed via alcohol dehydrogenation on basic sites (adsorbed O^−^, O_2_^−^ and pyrone-like structure), olefins are the products of dehydration process [12,20] and ethers are formed on acidic sites (carboxylic, phenolic and lactone groups). The CNM defectiveness [11,13,18], the content of carboxyl or other functional groups [15,16], heteroatoms (O, S, N, P) incorporated into the carbon structure [2,20] and unpaired electrons stimulating O^−^ and O_2_^−^ adsorption [22,23] affect the catalyst activity. However, the maximum alcohols conversion is usually observed over the catalysts with the highest surface acidity [20].

CNM oxidation with liquid HNO_3_ [24] or its vapors [25,26] increases acidity and defect density on the carbon surface. Gas phase oxidation is a more convenient oxidation route because it eliminates filtration, washing and drying steps. Moreover, the oxidation in the gas-phase can be considered as a suitable method for the modification of 3D carbon materials consolidated by spark plasma sintering (SPS) without compromising the pellet integrity [26].

Different CNMs were consolidated by SPS [26,27,28]. Consolidated CNMs can be used as adsorbents [29], biocompatible materials [30], catalysts, catalysts supports [31] and electrodes in energy storage devices [32].

Present work analyzes how the changes in the dimension of CNM materials from 0D to 3D affect the catalytic conversion of propanol-2 and butanol-2. We also propose a new method for the surface modification of 3D CNTs to increase their catalytic activity.

In this work we consolidate CNT into a 3D material by SPS, oxidize them with HNO_3_ vapors without pellets destruction and establish the correlations between the composition, structure and defectiveness that introduces paramagnetic centers. The sintered and oxidized CNT pellets were tested in secondary alcohols conversion to demonstrate the role of carbon structure dimension and surface activation method in the catalyst performance.

## 2. Materials and Methods

### 2.1. CNT Synthesis, Sintering and Gas-Phase Oxidation

CNTs were synthesized by pyrolytic decomposition of hexane over the Co/Mo-MgO catalyst at 750 °C for 30 min [33]. The synthesized material was refluxed with HCl solution to remove growth catalyst, washed with distilled water and dried in air at 120 °C. Amorphous carbon was removed by annealing of CNTs in air at 400 °C for 4 h. The produced material was named as CNTs_raw.

SPS was performed on a Labox-625 (Sinterland, Nagaoka City, Japan) apparatus at 1000–1100 °C for 5 min under the axial pressure of 22–30 MPa in a vacuum of 10^−2^ torr as described in [26,34]. Linear shrinkage of the specimen during SPS was continuously monitored. The heating speed was 100 °C·min^−1^. A graphite die set of 10 mm in diameter was used for the experiments. After sintering the pellets (named as CNTs_SPS) were separated from the die.

CNTs_SPS was oxidized with HNO_3_ vapors according to the procedure described in [25,26]. For this purpose, a pellet was placed in an open box with boiling HNO_3_ for 3 or 6 h. The oxidized samples were named as CNTs_SPS_x where x = 3, 6 is the oxidation time in hours.

### 2.2. Materials Characterization

The specific surface area was measured by low temperature nitrogen physisorption on an AUTOSORB-1C/MS/TPR analyzer (Quantachrome, Boynton Beach, FL, USA). High resolution transmission electron microscopy images were recorded on a JEOL 2100 F/Cs (Jeol, Akishima, Japan) microscope operated at 200 kV and equipped with a UHR pole tip, a spherical aberration corrector (CEOS, Heidelberg, Germany) and an EEL spectrometer (Gatan, Pleasanton, CA, USA).

X-ray photoelectron spectroscopy (XPS) was used to determine the surface composition, content and chemical state of oxygen and carbon atoms. The spectra were recorded on an Axis Ultra DLD instrument (Kratos Analytical, Manchester, UK) using monochromatic Al*K*_α_ radiation (1486.7 eV). Survey XPS spectra (Appendix A) were recorded with a pass energy of 160 eV, while a pass of 40 eV was used for high-resolution scans.

Raman spectra were registered on a LabRam HR800 UV (Horiba, Kyoto, Japan) spectrometer equipped with 5 mW argon laser (514.5 nm). Each sample was analysed in at least 5 points and the calculated parameters were averaged.

Continuous-wave electron paramagnetic resonance (EPR) measurements were performed at 9.70–9.90 GHz (X-band) using a BRUKER EMX 6/1 spectrometer. Spectra were recorded at microwave power of 0.635 mW, modulation amplitude of 1.0 G, time constant of 40.960 ms and receiver gain of 1.78 × 10^3^. The spin numbers (*N*) and g-values were measured using TEMPO (2,2,6,6-tetramethyl-1-piperidinyloxyl) in toluene solution as a standard sample with *g*_st_ = 2.0061 ± 0.0004 and *N*_st_ = 2.0792 × 10^16^ spins. The number of spins (*N_i_*) in the sample (*i*) was deduced using the expression *N_i_* = (*DI_i_* × *N*_st_)/*DI*_st_, where *DI_i_* and *DI*_st_ are, respectively, the double integrated intensities of the resonance curve of the sample and standard normalized by mass.

### 2.3. Catalytic Tests

The CNT samples were tested in catalytic conversions of propanol-2 and butanol-2 (99%) in a microreactor setup using the impulse method [18,35]. About 70 mg of the catalyst was placed between two layers of glass wool in the isothermal zone of a quartz tubular reactor (0.5 cm internal diameter and 14.5 cm length). The height of the catalyst layer depended on its density and was 0.5–2.5 cm. The error in the reaction temperature measurement was ±1 °C. Alcohol vapors (~5 mM) were injected with a syringe into the catalytic reactor fed with a nitrogen flow of 40 mL·min^−1^. The injected volume depended on the equilibrium pressure of the alcohol at room temperature. A mixture of products and unreacted alcohols was fed into a 1.3 m long and 0.5 cm in diameter column filled with Porapak-N sorbent and connected to the reactor. The column temperature was 130 °C. Alcohol conversion products were analyzed on a Chrom-5 chromatograph with a flame ionization detector.

The catalyst conversion was expressed as a percent of alcohol transformation. The selectivity of a product was calculated as a ratio of this product to the content of all products according to the following equation:(1)$$S=\frac{\mathrm{Pa}}{\sum^{\;}\mathrm{Pn}}.$$

The effective activation energy E_a_ was calculated as a tangent of the slope angle in the relationship diagram between 1/T and ln[ln(1/{1 − y})], where y is the alcohol conversion, as proposed in [36].

## 3. Results

### 3.1. CNT Characterization

Synthesized CNTs were of 15–20 nm in diameter and consisted of 10–15 carbon layers (Figure 1a). The internal channel diameter was 5–10 nm. SPS did not destroy the tubular CNT structure (Figure 1b). It was previously shown that CNTs could sinter at high temperature and/or pressure of SPS via graphene sheet formation. These graphene sheets link two individual CNTs [34,37,38]. Under less severe SPS conditions the improvements in the CNT structure were observed [34,39,40]. We believe that SPS conditions selected in our work (1000 °C and 22 MPa) were sufficient for the CNT surface recrystallization and defects «healing». However, at 1100 °C and 30 MPa the graphene sheets formation should be more pronounced. A large number of defects reappeared on the CNT surface after gas-phase oxidation (Figure 1c,d). The damage degree of the CNT structure dramatically increased with increasing the oxidation time from 3 up to 6 h. Surface layers in CNTs fragmentated after oxidation for 3 h (Figure 1c) and became corrugated after 6 h (Figure 1d). In general, the changes in the morphology of consolidated CNTs under gas-phase oxidation are similar to the evolution of the structure of non-consolidated CNTs under liquid-phase oxidation with HNO_3_ [41].

The low-temperature nitrogen adsorption-desorption isotherms of the samples (Figure 2) are of type II according to the IUPAC classification [42], which testifies to their macroporous nature. The calculated specific surface areas of CNTs_raw, CNTs_SPS and CNTs_SPS_3 are 204, 178 and 169 m^2^·g^−1^.

According to the XPS data the oxygen content was 1.0 at. % in CNTs_raw and 0.1 at. % in CNTs_SPS (Appendix A) and increased up to 11.3 and 14.9 at. % after oxidation for 3 to 6 h, respectively. The components in the high-resolution XPS spectra (Figure 3a,b) were attributed based on our previous works [24,43] and the XPS data for polymers [44]. The O1s spectra (Figure 3a) can be fitted with three components at 531.0, 532.0 and 533.1 eV. The former and the latter components can be respectively assigned to the double- and single bonded oxygen in carboxylic groups. The component at 532.0 eV corresponds to oxygen in hydroxylic groups and different NO_X_ species. (Appendix A).

The C1s spectra of oxidized samples (Figure 3b) mainly show the contribution from the asymmetric component at 284.3 eV attributed to *sp*^2^-carbon. The symmetric components at about 284.9, 286.6 and 288.7 eV can be assigned to *sp*^3^ C–C bonds, C–O and O=C–O species respectively (Appendix A).

Raman spectra of CNTs typically show three main bands: the D1 band (1360 cm^−1^, radial-breathing mode *A_lg_*), G band (1581 cm^−1^, *E*_2*g*_ stretching vibrations of C_6_ rings in the graphene plane) and 2D1 band (2700 cm^−1^, overtone of D1) [45]. Figure 4a shows the normalized Raman spectra of the samples. The increased *I_D_*_1_/*I_G_* ratio for the oxidized samples (Appendix A) reflects the more defective structure of graphene layers in oxidized CNTs.

Previous studies have shown the decrease in the *I_D_*_1_*/I_G_* ratio with increasing the SPS temperature or pressure in Raman spectra of consolidated CNTs because of the healing of defects during SPS, which improved the perfection of the graphitic structure [34]. In our case the *I_D_*_1_/*I_G_* ratio linearly increased with increasing O content under oxidation of sintered CNTs, while the *I*_2*D*1_/*I_G_* ratio showed the opposite trend (Figure 4b), which agrees with the previous results [46,47]. A new (D2) band appeared at 1600–1620 cm^−1^ in the Raman spectra of oxidized samples. Typically, the intensity of this band directly correlates with the intensity of the D1 band. However, the D2 line is excited only for surface layer and its intensity increases with the increase in the number of edge carbon atoms [46].

The EPR spectra of CNTs_SPS_3 and CNTs_SPS_6 samples show a strong paramagnetic response, while CNT_raw and CNTs_SPS are EPR silent (Figure 5). This means that paramagnetic centers in the former samples originated from unpaired electrons introduced by the oxidation process. The number of EPR centers in CNTs_SPS and CNTs_SPS_raw with low oxygen content are almost zero but it significantly increases under oxidation (CNTs_SPS_3 and CNTs_SPS_6). The absence of the EPR signal of CNTs_raw containing 1.5 at. % of oxygen could be explain by the lack of unpaired electrons in this material.

The experimental derivative EPR spectra of CNTs_SPS_3 and CNTs_SPS_6 (Figure 5a,c) were integrated to produce the absorption spectra (Figure 5b,d). These absorption spectra show different intensities (*DI*), widths (*W*) and positions (*g*-factor) and are well fitted with two Lorentzian lines. The fitting parameters of the absorption lines, calculated *g*-factors and numbers of spins (*N*) for CNTs_SPS_3 and CNTs_SPS_6 are presented in Table 1. The spin density associated with for both the narrow and broad lines increases after oxidation but the attribution of these lines is questionable. Based on the temperature dependence of EPR response in oxidized graphene, the broad line was earlier associated with the spin interaction between delocalized (mobile) electrons and localized π-electrons trapped in the extended aromatic structure [48,49,50,51,52]. At the same time, the wide and narrow lines were respectively assigned to localized and mobile electrons [53]. The intensities of both these lines increased with the oxidation time reflecting the increase in the number of paramagnetic centers, which correlates with the oxygen content and catalyst performance (see below). The different *g*-factors of the narrow and broad lines reflect the different surroundings of paramagnetic centers attributed to these lines.

### 3.2. Catalytic Tests

Conversions of propanol-2 and butanol-2 and products selectivities depend on the nature of catalysts and reaction temperature (Table 2). Alcohol conversion over CNTs_SPS was negligible. In the case of CNTs_SPS_3 conversions of both alcohols increased with temperature and achieved 100% at about 250 °C (Figure 6a,b) and shifted to low temperatures relative to CNTs_raw (Table 2). The main products of the propanol-2 conversion were propene, acetone and ether. The propene yield increased with temperature (Figure 6a) because of the different activation energies of two reaction pathways—dehydration and dehydrogenation [14,54].

The presence of the basic sites (carbonyl groups and adsorbed O^−^/O_2_^−^ [5,10,16,17,22]) on the CNTs_SPS_3 can be proved by approximately the same amount of acetone among the products at all studied temperatures [20,58]. Propanol-2 dehydrates to propylene and to diisopropyl over acid sites (usually carboxylic, lactonic and phenolic groups [16]) [20,58]. Therefore, CNTs_SPS_3 contained mostly acid sites, which were stable and did not eliminate from the CNTs_SPS_3 surface in the studied temperature range, which was established in [24].

CNTs_SPS_3 showed 100% selectivity to butene in the conversion of butanol-2 in all temperature range (Figure 6b). We can conclude that carboxylic groups are the main catalytic active sites, whereas the amount of aldehyde and carbonyl groups in this sample is negligible. It should be noted that the oxidation of CNMs always leads to selectivity to butene (Table 2). Primary alcohols are converted at higher temperatures. For example, 100% conversion of *n*-butanol-1 was observed only at 330 °C and the products were propion and butyr aldehydes, butane and butyl butyrate [14]. This can be probably explained by the different alcohol polarization and, therefore, different sorption mechanism [59].

The calculated activation energies point to the internal diffusion control of the reaction at higher temperatures (Figure 6c,d). This means that the limiting stage of the conversion of both alcohols is their diffusion in pores. However, at 160 °C the conversion of butanol-2 over CNTs_SPS_3 become kinetically controlled that means low velocity of alcohol conversion reaction itself (Figure 6d).

## 4. Discussion

In the catalytic conversion of alcohols the nature of both the substrate (primary, secondary or tertiary alcohol) and catalyst are important. In the present discussion, we try to estimate the role of the dimension of CNMs and possible ways for improving the catalytic activity of these materials.

CNMs of different dimensions were tested in catalytic conversion of aliphatic alcohols (Table 2). Firstly, the catalytic activity strongly depends on the functionalization degree of the CNM surface: conversion increases and temperature of 50% conversion decreases with increasing the oxygen content. Secondly, the type of hybridization of carbon atoms (*sp*^2^ or *sp*^3^) also affects the catalytic activity. Terminal *sp*^3^-carbon atoms on the surface of 0D NDs are intrinsic «ready to use» catalytically active sites because of adsorbed water, oxygen. And other molecules on the surface as it was shown in [5,18]. At the same time, onion-like carbon nanoparticles with the same zero dimension and *sp*^2^-hibridized C atoms are less active and selective in the butalol-1 conversion [14].

CNTs and carbon nanofibers are 1D carbon materials. *sp*^2^-hybridisized carbon atoms in their structure are predominantly rolled in tubes. This results in the partial delocalization of the aromatic π-electron system [5,60]. According to first-principles calculations defect sites in bent graphene units are active for water dissociation [61]. The catalytic activity can also originate from heteroatoms [3] and defects in carbon matrix [13,17,55,62] because of the formation of Stone-Wales defects.

Graphene oxide (GO) and graphene nanoflakes (GNFs) are attributed to 2D carbon materials. GNFs can be considered as a flat polyhedra with 5–15 carbon layers and bent edges [63]. GO and CNFs are inactive in alcohol conversion (Table 2), [11,14] though they contain a small amount of oxygen (Table 2). Probably oxygen in these materials is bonded in catalytically inactive −OH groups. GNFs became catalytically active only after oxidation (Table 2, Appendix A).

The catalytic conversion of alcohols over oxidized GNFs is less efficient than over oxidized CNTs despite the higher oxygen content in the former: the alcohol conversions at 250 °C are respectively 100% and 50% (Table 2, Appendix A). We can propose that the oxidation of GNFs affects their structure to a less degree producing only the surface oxygen species.

CNTs sintered by SPS can be classified as a 3D materials [33,34]. According to the Raman spectroscopy data the structure of carbon layers in CNTs improves under SPS consolidation (Figure 4). The EPR spectroscopy did not reveal unpaired electrons in sintered 3D CNTs. Therefore, their structure does not *a priori* contain active centers for catalytic conversion of aliphatic alcohols. At the same time, varying the sintering conditions CNTs can be transformed into other carbon phases [33,34,37] but this requires further investigation.

Oxidation of the sintered 3D CNTs significantly enhanced their catalytic activity via introduction of functional groups, increase in defectiveness and appearance of unpaired electrons. At the same time, the specific surface area of oxidized sintered CNTs is lower than those of raw and consolidated CNTs. Therefore, it cannot contribute to the enhanced catalytical efficiency of the oxidized 3D CNTs. The activity of this material is close to that of activated carbon—another 3D carbon material (Table 2), [15,16,20]. Our work demonstrates a new way for producing a rigid 3D CNT material with the desirable characteristics.

## 5. Conclusions

Consolidation of CNTs by SPS decreased their defectiveness via surface recrystallization. The oxidation of consolidated CNT pellets with nitric acid vapors preserved their rigid structure. The oxygen content in the oxidized samples increased from 11.3 up to 14.9 at. % with increasing the oxidation time from 3 to 6 h. The defectiveness of these samples simultaneously increased as determined by Raman spectroscopy.

Sintered CNTs lost their catalytic activities in the conversion of second aliphatic alcohols because of the elimination of defects and functional groups on their surface. The subsequent oxidation of the sintered CNTs restored the catalytic activity and significantly enhanced it compared to unsintered ones. The high activity of the oxidized sintered CNTs resulted from the high oxygen content as measured by XPS and large number of unpaired electrons as detected by EPR spectroscopy. High content of carboxylic groups on the surface provided high selectivity of the propanol-2 (butanol-2) conversion to propene (butene).

The structure of CNM affects the content and/or ratio of *sp*^2^ and *sp*^3^-hybridized carbon atoms in the material. The experimental and literature data demonstrate that *sp*^3^-hybridized carbon atoms on the surface are probably the preferable site for catalytic conversion of alcohols.

## Figures and Tables

**Figure 1 nanomaterials-11-00352-f001:**
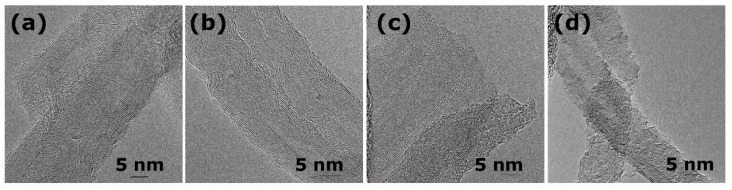
Transmission electron microscopy (TEM) images of CNTS_raw (**a**), CNTs_SPS (**b**), CNTs_SPS_3 (**c**), CNTs_SPS_6 (**d**) demonstrate the destruction of carbon nanotube (CNT) structure during oxidation.

**Figure 2 nanomaterials-11-00352-f002:**
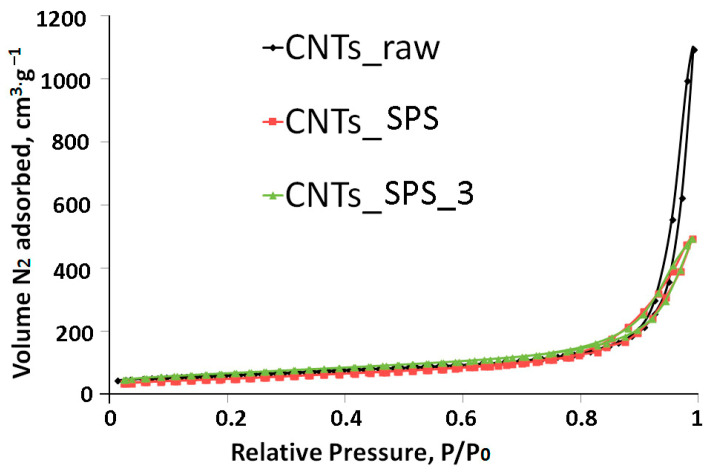
Low-temperature nitrogen physisorption isotherms of CNTs samples. All isotherms are typical for macroporous solids.

**Figure 3 nanomaterials-11-00352-f003:**
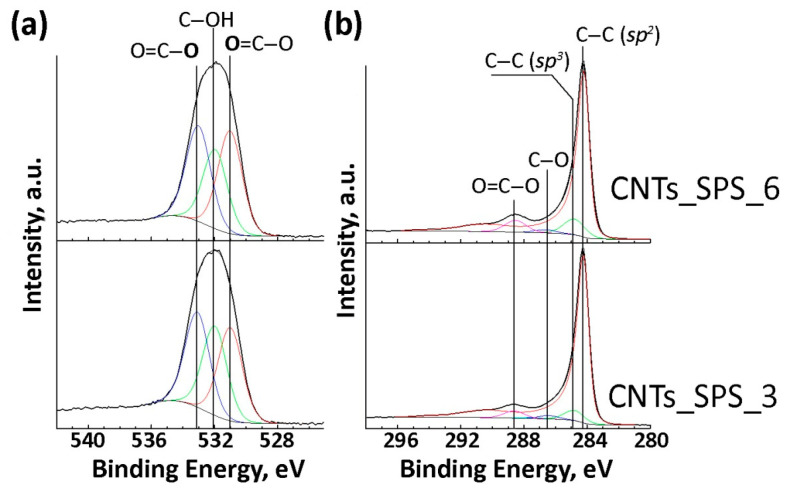
O1s (**a**) and C1s (**b**) X-ray photoelectron spectroscopy (XPS) spectra of oxidized sintered CNTs.

**Figure 4 nanomaterials-11-00352-f004:**
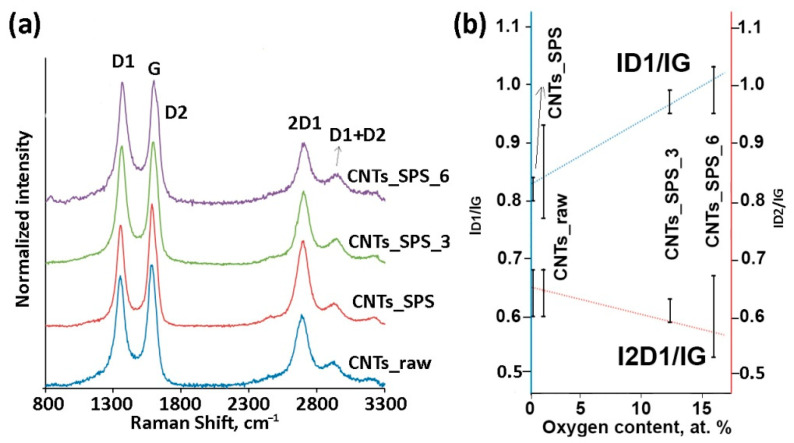
Normalized Raman spectra of CNT samples (**a**) and correlation between *I_D_*/*I_G_* ratios and XPS oxygen content (**b**). The increase in *I_D_*_1_/*I_G_* and decrease in *I*_2*D*1_/*I_G_* ratios under oxidation indicates the increase in CNT defectiveness.

**Figure 5 nanomaterials-11-00352-f005:**
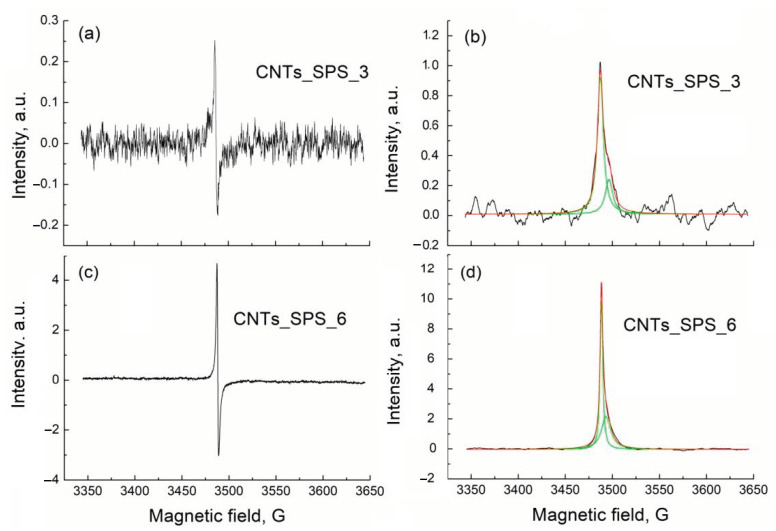
First-derivative (**a**,**c**) and absorption (**b**,**d**) electron paramagnetic resonance (EPR) spectra of CNTs_SPS_3 and CNTs_SPS_6. Background signal was subtracted using the Bruker WinEPR Processing Software. Absorption spectra were fitted with two Lorentzian lines.

**Figure 6 nanomaterials-11-00352-f006:**
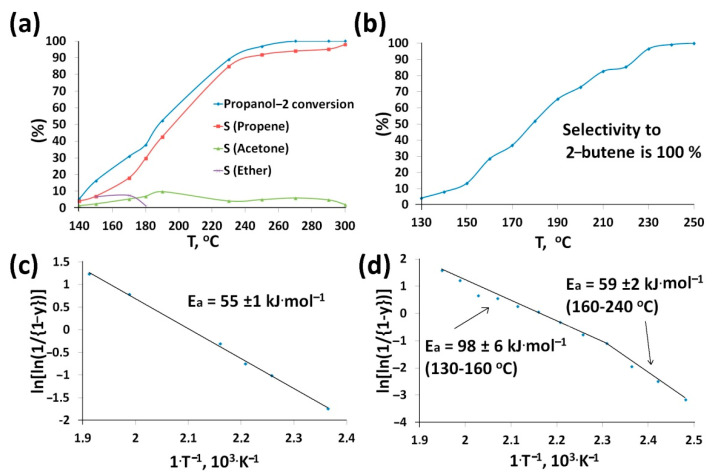
Temperature dependences of propanol-2 (**a**) and butanol-2 (**b**) conversions and products selectivities over CNTs_SPS_3. The activation energies of propanol-2 (**c**) and butanol-2 (**d**) conversion were expressed in the Habgood-Bassett coordinates according to [36], where y is the fractional conversion.

**Table 1 nanomaterials-11-00352-t001:** Line widths (*W*), g-factors and numbers of spins (*N*) calculated from EPR spectra of CNTs_SPS_3 and CNTs_SPS_6. Subscript n and b denote narrow and broad lines. Typical error in determination of spin number is less than 15%.

Sample	Width (G)		*g*-Factor		Number of Spins (*N*) (10^16^ (Spin·g^−1^))	
	*W* _n_	*W* _b_	*g* _n_	*g* _b_	*N* _n_	*N* _b_
CNTs_SPS_3	7.5 ± 0.2	9.5 ± 0.7	2.006 (1)	2.001 (1)	1.5 (1)	0.47 (7)
CNTs_SPS_6	3.34 ± 0.03	11.77 ± 0.12	2.0060 (8)	2.0032 (8)	7.0 (5)	5.5 (5)

**Table 2 nanomaterials-11-00352-t002:** Oxygen content in carbon nanomaterial (CNM) materials of different dimensions (D) and their catalytic properties in propanol-2 and butanol-2 conversion. T_50_—temperature of 50% conversion; S—selectivity; E_a_—activation energy. NC, LC and HC denote respectively negligible, lower than 50% or higher than 50% conversion in all studied temperature range.

D	CNMs	O (at. %)	Propanol-2				Butanol-2				Ref.
			T_50_ (°C)	S (%)		E_a_, (kJ·mol^−1^)	T_50_ (°C)	S (%)		E_a_ (kJ·mol^−1^)	
				Propene	Acetone			Butene	Butanone		
0D	ND	7.4	240	22	78	87 ± 2	–	–	–	–	[55]
1D	Onion-like carbon	n/a					LC	50	40		[14] ^a^
	CNTs_raw ^b^	0.64	250	40	5	74 ± 1	280	58	22	45 ± 1	[13,55]
	Oxidized CNTs_raw	1.4	190	77	23	59 ± 2	HC	100	0	36 ± 1	
	Oxidized CNTs_cyl_ ^b^	9.4	250								
	CNTs_con_ ^c^	2.3	LC	14	86	98 ± 3	LC	34	66	86 ± 4	[13]
	Oxidized CNTs_con_ ^c^	9.4	HC	30	0	54 ± 1	190	100	0	–	[13,55]
2D	CNFs	1.8	NC				NC				[13]
	Oxidized CNFs ^d^	22.1	240	95	5	75 ± 2	270	100		37 ± 2 62 ± 1	Present work, Appendix A
	Graphene oxide						LC	45	40		[14] ^a^
3D	Activated carbon (H_3_PO_4_, 450 °C)	23.8	175	100		98.5					[20]
	CNTs_SPS	0.1	NC				NC				Present work
	CNTs_SPS_3	11.3	190	98	2	55 ± 1	180	100	0	59 ± 2 98 ± 6	Present work

^a^ Butanol-1 conversion was studied in this work. ^b^ CNTs_cyl_—CNTs with the cylindrical structure of layers. ^c^ CNTs_con_—CNTs with the conical structure of layers. ^d^ Synthesis and characterization of the sample were described earlier [56,57].

## Data Availability

The data presented in this study are available on request from the corresponding author.

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
