# Peer review of "Conversion of Secondary C3-C4 Aliphatic Alcohols on Carbon Nanotubes Consolidated by Spark Plasma Sintering"

_nanomaterials, 2021, doi:10.3390/nano11020352_

Round 1

Reviewer 1 Report

This is a thorough study, with well thought and well supported conclusions, and a good read.

Please consider the following corrections:

  • Your abstract does not do justice to your paper. It starts as a lab report summary and ends up as an introduction. It should start by introducing the general field and general scientific question, explain the hypothesis and briefly the methodology. Summary the results, the conclusions and their implications (don't say they are discussed....say what the conclusion of the discussion is!). See, for example, https://www.wiley.com/network/researchers/preparing-your-article/how-to-write-a-scientific-abstract
  • similarly your figure captions are not descriptive. They should summarise the scientific conclusion that the figures convey, with the same principle as the abstract, that they should be 'self-contained' and be able to be understood on their own, without reference to the text. See, for example: https://chec.engineering.cornell.edu/visuals/captions-for-figures-in-documents/
  • There are some typos: line 26, page 1: "..can be tuneD in a wider range BY varying...", Page 9 Line 274 Stone-WALES defects, P9 line 261 "depends on the FUNCTIONALIZATION degree.."
  • P2 line 87, Gatan is a US company. https://www.gatan.com/company/about-gatan
  • p4 lines 154-161 please give reference(s) for the tabulated peak assignations to the specific atomic bonds.
  • p4 line 166. the conclusion that "...reflects THE more defective structure..." ties in with TEM observation, so the authors may wish to mention the connection.
  • P4 line 167 "Earlier.." it would be best to rephrase "previous studies have shown...?" as it can be confusing.
  • p5 line 174 "exCited"?
  • P6 Table 1, please provide confidence margins (std deviations)
  • P7 lines 242-244. please provide reference for this statement.
  • p9 - would have been good to measure the surface area of the materials (from raw to pellet to functionalised), to determine/exclude a pore size (and surface area) contribution to the catalytic efficiency. Perhaps a short statement may be worth including here?
  • Overall, there are very few grammatical mistakes, but i would urge the use of a grammar and syntax checker (not just spellcheck) as there are numerous commas missing - either Word's built-in one or apps such as Grammarly.

Author Response

  • Your abstract does not do justice to your paper. It starts as a lab report summary and ends up as an introduction. It should start by introducing the general field and general scientific question, explain the hypothesis and briefly the methodology. Summary the results, the conclusions and their implications (don't say they are discussed....say what the conclusion of the discussion is!).

Reply:

Abstract has been re-written according to the reviewer’s recommendations.

  • similarly your figure captions are not descriptive. They should summarise the scientific conclusion that the figures convey, with the same principle as the abstract, that they should be 'self-contained' and be able to be understood on their own, without reference to the text.

Reply:

More detailed figure captions have been provided.

  • There are some typos: line 26, page 1: "..can be tuneD in a wider range BY varying...", Page 9 Line 274 Stone-WALES defects, P9 line 261 "depends on the FUNCTIONALIZATION degree.."
  • P2 line 87, Gatan is a US company. https://www.gatan.com/company/about-gatan

Reply:

All necessary corrections have been made.

  • p4 lines 154-161 please give reference(s) for the tabulated peak assignations to the specific atomic bonds.

Reply

The references have been added.

  • p4 line 166. the conclusion that "...reflects THE more defective structure..." ties in with TEM observation, so the authors may wish to mention the connection.
  • P4 line 167 "Earlier.." it would be best to rephrase "previous studies have shown...?" as it can be confusing.
  • p5 line 174 "exCited"?

Reply:

All necessary corrections have been made.

  • P6 Table 1, please provide confidence margins (std deviations)

Reply:

The confidence margins have been added to Table 1.

  • P7 lines 242-244. Please provide reference for this statement.

Reply

The reference has been added.

  • p9 - would have been good to measure the surface area of the materials (from raw to pellet to functionalised), to determine/exclude a pore size (and surface area) contribution to the catalytic efficiency. Perhaps a short statement may be worth including here?

Reply:

Nitrogen physisorption isotherms and SBET values are presented in p. 4. The macroporous nature of the isotherms doesn’t allow pore size analysis. At the same time the oxidized material shows the lowest specific surface area. To emphasize this fact the following statement has been added to the discussion part (p. 9): “At the same time, the specific surface area of oxidized sintered CNTs is lower than those of raw and consolidated CNTs. Therefore, it can’t contribute to the enhanced catalytical efficiency of the oxidized material.”

  • Overall, there are very few grammatical mistakes, but i would urge the use of a grammar and syntax checker (not just spellcheck) as there are numerous commas missing - either Word's built-in one or apps such as Grammarly.

Reply:

The manuscript has been thoroughly re-checked and corrected.

Reviewer 2 Report

The authors present a study of carbon nanotubes produced using a spark plasma sintering process.  The paper presents results from Raman, EPR, and XPS analysis of processed CNTs. The paper provides findings that are of interest to the community. 

A few suggestions:

Line 26, change “tune” to “tuned”

Line 31 change “alcohols” to “alcohol”

Line 38 change “alcohols” to “alcohol”

Line 48 change “defectiveness of” to “defect density on”

Line 133 change “enough” to “sufficient”

Line 174 should “exited” be “excited” or “exists”?

Author Response

A few suggestions:

Line 26, change “tune” to “tuned”

Line 31 change “alcohols” to “alcohol”

Line 38 change “alcohols” to “alcohol”

Line 48 change “defectiveness of” to “defect density on”

Line 133 change “enough” to “sufficient”

Line 174 should “exited” be “excited” or “exists”?

Reply:

All necessary corrections have been made.